# Pre-training Tasks for Embedding-based Large-scale Retrieval

**Wei-Cheng Chang**[*]**, Felix X. Yu, Yin-Wen Chang, Yiming Yang, Sanjiv Kumar**
Carnegie Mellon University & Google
{wchang2,yiming}@cs.cmu.edu, {felixyu,yinwen,sanjivk}@google.com

## Abstract

We consider the large-scale query-document retrieval problem: given a query (e.g., a question), return the set of relevant documents (e.g., paragraphs containing the answer) from a large document corpus. This problem is often solved in two steps. The retrieval phase first reduces the solution space, returning a subset of candidate documents. The scoring phase then re-ranks the documents. Critically, the retrieval algorithm not only desires high recall but also requires to be highly efficient, returning candidates in time sublinear to the number of documents. Unlike the scoring phase witnessing significant advances recently due to the BERT-style pre-training tasks on cross-attention models, the retrieval phase remains less well studied. Most previous works rely on classic Information Retrieval (IR) methods such as BM-25 (token matching + TF-IDF weights). These models only accept sparse handcrafted features and can not be optimized for different downstream tasks of interest. In this paper, we conduct a comprehensive study on the embedding-based retrieval models. We show that the key ingredient of learning a strong embedding-based Transformer model is the set of pre-training tasks. With adequately designed paragraph-level pre-training tasks, the Transformer models can remarkably improve over the widely-used BM-25 as well as embedding models without Transformers. The paragraph-level pre-training tasks we studied are Inverse Cloze Task (ICT), Body First Selection (BFS), Wiki Link Prediction (WLP), and the combination of all three.

## 1 Introduction

We consider the large-scale retrieval problem: given a query, return the most relevant documents from a large corpus, where the size of the corpus can be hundreds of thousands or more. One can view this problem as learning a scoring function $f : \mathcal{X} \times \mathcal{Y} \to \mathbb{R}$, that maps a pair of a query and a document $(\boldsymbol{q}, \boldsymbol{d}) \in \mathcal{X} \times \mathcal{Y}$ to a score $f(\boldsymbol{q}, \boldsymbol{d})$. The function should be designed such that the relevant $(\boldsymbol{q}, \boldsymbol{d})$ pairs have high scores, whereas the irrelevant ones have low scores. Many real-world applications besides query-document retrieval can be cast into this form. For example, in recommendation systems, $\boldsymbol{q}$ represents a user query and $\boldsymbol{d}$ represents a candidate item to recommend (Krichene et al., 2019). In extreme multi-label classification, $\boldsymbol{q}$ represents a web-page document and $\boldsymbol{d}$ represents the categories or hashtags of interests (Jain et al., 2019; Chang et al., 2019). In open-domain question answering, $\boldsymbol{q}$ represents a question and $\boldsymbol{d}$ represents an evidence passage containing the answer (Chen et al., 2017; Hu et al., 2019; Lee et al., 2019).

Central to the above is designing the scoring function $f$. Recently, BERT (Devlin et al., 2019), along with its many successors such as XLNet (Yang et al., 2019b) and RoBERTa (Liu et al., 2019), has led to significant improvements to many NLP tasks such as sentence pairs classification and question-answering. In BERT, the scoring function $f$ is a pre-trained deep bidirectional Transformer model. While BERT-style cross-attention models are very successful, it cannot be directly applied to large-scale retrieval problems because computing $f(\boldsymbol{q}, \boldsymbol{d})$ for every possible document can be prohibitively expensive. Thus, one typically first uses a less powerful but more efficient algorithm (another scoring function $f$) to reduce the solution space (the "retrieval phase"), and then use the BERT-style model to re-rank the retrieved documents (the "scoring phase").

---

[*]work performed when interning at Google.

The retrieval phase is critical. Ideally speaking, the algorithm should have a high recall; otherwise, many relevant documents won't even be considered in the scoring phase. The algorithm also needs to be highly efficient: it should return a small subset of relevant documents in time sublinear to the number of all documents. Although significant developments are advancing the scoring algorithms, the retrieval algorithms remain less studied, and this is the focus of this paper.

The retrieval algorithm can be put into two categories. The first type is classic information retrieval (IR) algorithms relying on token-based matching. One example is BM-25 (Robertson et al., 2009), which remains to be the most commonly-used (Nguyen et al., 2016; Yang et al., 2017; 2019a) and hard to beat (Chapelle & Chang, 2011; Lee et al., 2019) algorithm. Here the scoring function $f$ is based on token-matching between the two high-dimensional sparse vectors with TF-IDF token weights, and retrieval can be done in sublinear time using the inverted index. Despite the wide usage, these algorithms are handcrafted and therefore cannot be optimized for a specific task.

The second option is an embedding-based model that jointly embeds queries and documents in the same embedding space and use an inner product or cosine distance to measure the similarity between queries and documents. Let the query embedding model be $\phi(\cdot)$ and the document embedding model be $\psi(\cdot)$. The scoring function is

$$f(\boldsymbol{q}, \boldsymbol{d}) = \langle \phi(\boldsymbol{q}), \psi(\boldsymbol{d}) \rangle.$$

In the inference stage, retrieving relevant documents then becomes finding the nearest neighbors of a query in the embedding space. Since the embeddings of all candidate documents can be pre-computed and indexed, the inference can be done efficiently with approximate nearest neighbor search algorithms in the embedding space (Shrivastava & Li, 2014; Guo et al., 2016).

In this paper, we refer to the above embedding-based model as the *two-tower retrieval model*, because the query and document embeddings are coming from two separate "towers" of neural networks. In the literature, it is also known as the Siamese network (Das et al., 2016; Triantafillou et al., 2017) or dual-encoder model (Cer et al., 2018; Mazaré et al., 2018). Compared to the sparse token-based models, the two-tower models can capture deeper semantic relationships within queries and documents, and the models can be optimized specifically for the task being considered.

In the heart of two-tower models is the embedding functions $\phi(\cdot)$ and $\psi(\cdot)$. A modern choice is using Transformers to model the attention within queries and within documents, rather than the cross-attention between them as in the BERT model. The token-level masked-LM (MLM) pre-training task is crucial to the success of BERT-style cross-attention models. Nevertheless, what pre-training tasks are useful for improving two-tower Transformer models in large-scale retrieval, remains a crucial yet unsolved research problem. In this paper, we aim to answer this question by studying different pre-training tasks for the two-tower Transformer models. We contribute the following insight:

- The two-tower Transformer models with proper pre-training can significantly outperform the widely used BM-25 algorithm;

- Paragraph-level pre-training tasks such as Inverse Cloze Task (ICT), Body First Selection (BFS), and Wiki Link Prediction (WLP) hugely improve the retrieval quality, whereas the most widely used pre-training task (the token-level masked-LM) gives only marginal gains.

- The two-tower models with deep transformer encoders benefit more from paragraph-level pre-training compared to its shallow bag-of-word counterpart (BoW-MLP).

To the best of our knowledge, this is the first comprehensive study on pre-training tasks for efficient large-scale retrieval algorithms. The rest of the paper is organized as follows. We start by introducing the two-tower retrieval model in Section 2. The pre-training tasks are presented in 3, and the experiments and analysis are presented in Section 4. Finally, we conclude this work in Section 5.

## 2 THE TWO-TOWER RETRIEVAL MODEL

Given a query $\boldsymbol{q} \in \mathcal{X}$ and a document $\boldsymbol{d} \in \mathcal{Y}$, we consider two-tower retrieval models that consist of two encoder functions, $\phi : \mathcal{X} \to \mathbb{R}^k$ and $\psi : \mathcal{Y} \to \mathbb{R}^k$ which map a sequence of tokens in $\mathcal{X}$ and $\mathcal{Y}$ to their associated embeddings $\phi(\boldsymbol{q})$ and $\psi(\boldsymbol{d})$, respectively. The scoring function $f : \mathbb{R}^k \times \mathbb{R}^k \to \mathbb{R}$

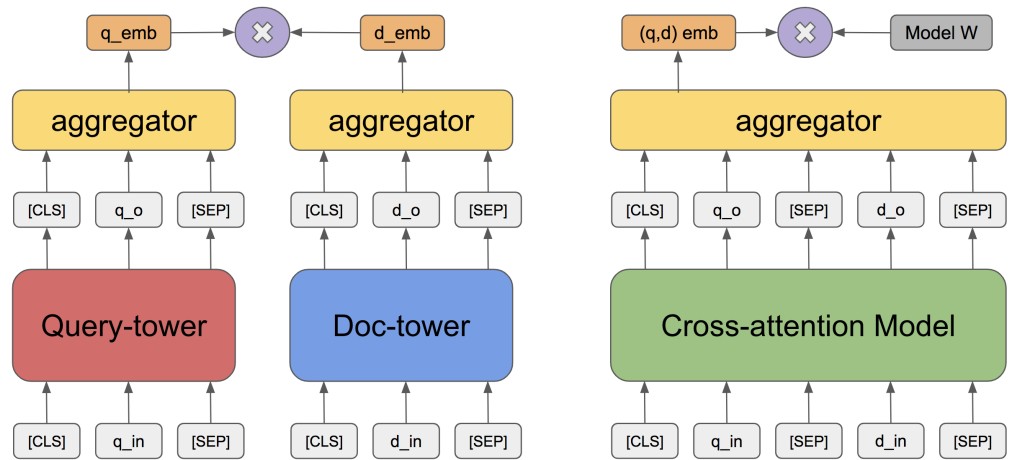

Figure 1: Difference between two-tower models and cross-attention models. Following previous works, we consider [CLS] embedding and average pooling as the aggregator's output for the two-tower Transformer model and the two-tower MLP model, respectively.

is then defined to be the inner product[1] of the embeddings

$$f(\boldsymbol{q}, \boldsymbol{d}) = \langle \phi(\boldsymbol{q}), \psi(\boldsymbol{d}) \rangle. \tag{1}$$

In this paper, we are interested in parameterizing the encoders $\phi, \psi$ as deep Transformer models (Vaswani et al., 2017) due to its expressive power in modeling natural language.

In the rest of this section, we illustrate the advantage of two-tower models in the inference phase; discuss the pros and cons of two-tower models in comparison with BERT-like cross-attention models; present the learning procedure of estimating model parameters under maximum likelihood principle; and review the related works.

**Inference** The difference between two-tower models and cross-attention models is shown in Figure 1. The advantage of two-tower models is the efficiency in the inference time. First, all the document embeddings can be pre-computed. Then, given an unseen query $\boldsymbol{q}$, we only need to rank the document based on its inner product with the query embedding. This is way more efficient than running inference on a cross-attention BERT-style model (often used in the scoring stage). To see this, the scoring function of BERT-style model is with the form

$$f_{\theta, \boldsymbol{w}}(\boldsymbol{q}, \boldsymbol{d}) = \psi_\theta(\boldsymbol{q} \oplus \boldsymbol{d})^T \boldsymbol{w}, \tag{2}$$

where $\oplus$ denotes the concatenate operation of the query and the document sequence and $\boldsymbol{w} \in \mathbb{R}^k$ is an additional model parameters. In BERT, for each query, one has to make the above expensive inference on all documents. For example, with the 128-dimensional embedding space, inner product between 1000 query embeddings with 1 million document embeddings only takes hundreds of milliseconds on CPUs, while computing the same scores with cross-attention models takes hours if not more even on GPUs.

Furthermore, retrieving the closest documents in the embedding space can be performed in sublinear time with the well-studied maximum inner product (MIPS) algorithms with almost no loss in recall (Shrivastava & Li, 2014; Guo et al., 2016).

**Learning** One unique advantage of the two-tower retrieval model in comparison with classic IR algorithms is the ability to train it for specific tasks. In this paper, we assume that the training data is presented as relevant "positive" query-document pairs $\mathcal{T} = \{(\boldsymbol{q}_i, \boldsymbol{d}_i)\}_{i=1}^{|\mathcal{T}|}$. Let $\theta$ be the model parameters. We estimate the model parameters by maximizing the log likelihood

---

[1]This also includes cosine similarity scoring functions when the embeddings $\phi(\boldsymbol{q}), \psi(\boldsymbol{d})$ are normalized.

$\max_\theta \sum_{(\boldsymbol{q},\boldsymbol{d}) \in \mathcal{T}} \log p_\theta(\boldsymbol{d}|\boldsymbol{q})$ where the conditional probability is defined by the Softmax:

$$p_\theta(\boldsymbol{d}|\boldsymbol{q}) = \frac{\exp\left(f_\theta(\boldsymbol{q},\boldsymbol{d})\right)}{\sum_{\boldsymbol{d}' \in \mathcal{D}} \exp\left(f_\theta(\boldsymbol{q},\boldsymbol{d}')\right)}, \tag{3}$$

and $\mathcal{D}$ is the set of all possible documents. The Softmax involves computing the expensive denominator of Equation (3), a.k.a, the partition function, that scales linearly to the number of documents. In practice, we use the Sampled Softmax, an approximation of the full-Softmax where we replace $\mathcal{D}$ by a small subset of documents in the current batch, with a proper correcting term to ensure the unbiasedness of the partition function (Bengio & Senécal, 2008). Sampled Softmax has been widely used in language modeling (Chen et al., 2016; Grave et al., 2017), recommendation systems (Yu et al., 2017; Krichene et al., 2019) and extreme classification (Blanc & Rendle, 2018; Reddi et al., 2019).

Since we often have a limited amount of supervised data from the downstream task, it is important to first train the retrieval model with positive pairs $\mathcal{T}$ from a set of pre-training tasks. We then fine-tune it with positive pairs $\mathcal{T}$ from the downstream task. We will present the set of pre-training tasks we study in Section 3.

**Related Works**  Cer et al. (2018) study the two-tower Transformer model as a universal sentence encoder. The model is learned with multiple tasks including the unsupervised Skip-Thought task (Kiros et al., 2015), the supervised conversation input-response task (Henderson et al., 2017), and the supervised sentence classification SNLI task (Bowman et al., 2015). Humeau et al. (2019) propose the Poly-encoders architecture to balance the computation/expressiveness tradeoff between two-tower models and cross-attention models. Reimers & Gurevych (2019) fine-tune the deep two-tower models on two supervised datasets, SNLI and MNLI (Williams et al., 2018), then apply it in solving other downstream tasks. Unlike all the above works that consider training the two-tower Transformer models on a limited amount of supervised corpus for the sentence classification tasks, we study different pre-training tasks and their contributions in the large-scale retrieval settings.

Another closely related topic is the open-domain question answering. Previous works consider using BM25 or other lexical matching methods to retrieve the top-k relevant passages efficiently and then deploy the more expensive cross-attention scoring function to find the answer (Chen et al., 2017; Yang et al., 2017; 2019a). Das et al. (2019) encode query and document separately with LSTM encoders. They employ a training procedure different from ours and do not consider pre-training. Very recently, Lee et al. (2019) propose to pre-train two-tower Transformer models with the Inverse Cloze Task (ICT) to replace BM25 in the passage retrieval phase. The advantage is that the retriever can be trained jointly with the reader/scorer. Nevertheless, their pre-trained two-tower models do not outperform BM25 on the SQuAD dataset, potentially because the fine-tuning is only performed on the query-tower.

Model distillation (Hinton et al., 2015) can be used to compress expensive BERT-like cross-attention models into efficient two-tower Transformer models for large-scale retrieval problems. For example, Tang et al. (2019) demonstrate initial success in distilling the BERT model into a two-tower model with BiLSTM as encoders. The pre-training tasks we study in this paper can be used as additional supervision in the distillation process, and therefore complementary to model distillation.

## 3 PRE-TRAINING TASKS OF DIFFERENT SEMANTIC GRANULARITIES

As mentioned in Section 2, due to the limited amount of supervised data from downstream tasks, a crucial step of learning deep retrieval models is to pre-train the model with a set of pre-training tasks (we will verify this in Section 4). Sentence-level pre-training tasks have been studied before. One example is reconstructing the surface form of surrounding sentences given the encoded sentence (Le & Mikolov, 2014; Kiros et al., 2015), and another one is discriminating the next sentence from random candidates (Jernite et al., 2017; Logeswaran & Lee, 2018).

In this paper, we assume that the pre-training data is defined as positive query-document $(\boldsymbol{q}, \boldsymbol{d})$ pairs. A good pre-training task should have the following two properties. **1)** It should be relevant to the downstream task. For example, when solving the question-answering retrieval problem, the model should capture different granularities of semantics between the query and document. The semantics

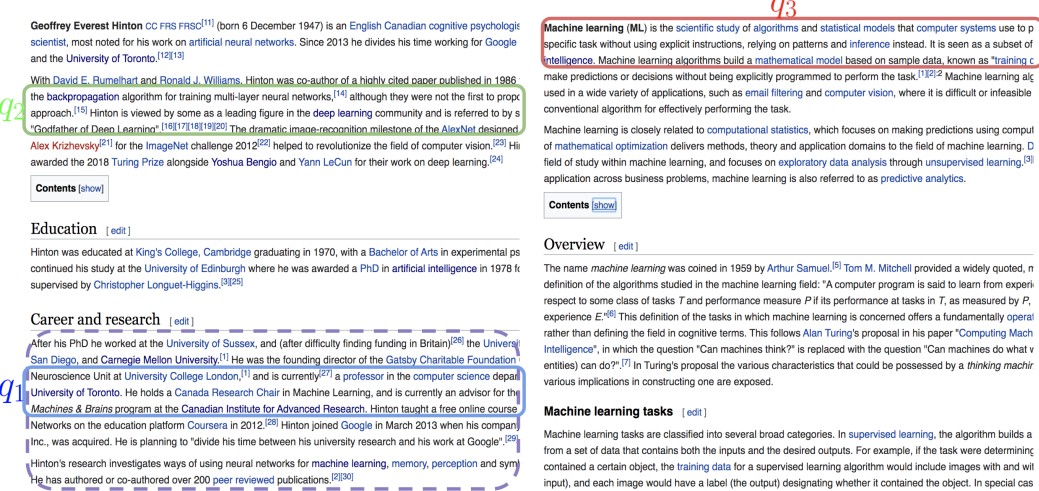

Figure 2: An illustrative example of the three pre-training tasks where each query $q$ is highlighted in different colors. All queries are paired with the same text block $d$. Concretely, $(q_1, d)$ of ICT is defined locally within a paragraph; $(q_2, d)$ of BFS is defined globally within an article; $(q_3, d)$ of WLP is defined distantly across two related articles hyper-linked by the Wikipedia entity.

can be the local context within a paragraph, global consistency within a document, and even semantic relation between two documents. **2)** It should be cost-efficient to collect the pre-training data, ideally not requiring additional human supervision.

In light of the above requirements, we present three pre-training tasks that emphasize different aspects of semantics between queries and documents: Inverse Cloze Task (ICT), Body First Selection (BFS), and Wiki Link Prediction (WLP). In specific, BFS and WLP are newly proposed in this paper. The training data for all these tasks can be freely obtained based from Wikipedia without an additional manual labeling process. Figure 2 provides illustrative examples of these tasks.

**Inverse Cloze Task (ICT)**    Given a passage $p$ consisting of $n$ sentences, $p = \{s_1, \ldots, s_n\}$, the query $q$ is a sentence randomly drawn from the passage, $q = s_i, i \sim [1, n]$, and the document $d$ is the rest of sentences, $d = \{s_1, \ldots, s_{i-1}, s_{i+1}, \ldots, s_n\}$. See $(q_1, d)$ in Figure 2 as an example. This task captures the semantic context of a sentence and was originally proposed by Lee et al. (2019).

**Body First Selection (BFS)**    We propose BFS to capture semantic relationship outside of the local paragraph. Here, the query $q_2$ is a random sentence in the first section of a Wikipedia page, and the document $d$ is a random passage from the same page (Figure 2). Since the first section of a Wikipedia article is often the description or summary of the whole page, we expect it to contain information central to the topic.

**Wiki Link Prediction (WLP)**    We propose WLP to capture inter-page semantic relation. The query $q_3$ is a random sentence in the first section of a Wikipedia page, and the document $d$ is a passage from another page where there is a hyperlink link to the page of $q_3$ (Figure 2). Intuitively, a hyperlink link indicates relationship between the two Wikipedia pages. Again, we take a sentence from the first section because it is often the description or summary of the topic.

**Masked LM (MLM)**    In addition to the above tasks, we also consider the classic masked language model (MLM) pre-training task as a baseline: predict the randomly masked tokens in a sentence. MLM is the primary pre-training task used in BERT (Devlin et al., 2019).

| Pre-training tasks | #tokens | #pairs | avg. #query tokens | #doc tokens |
|:---:|:---:|:---:|:---:|:---:|
| ICT | 11.2B | 50.2M | 30.41 | 193.89 |
| BFS | 3.3B | 17.5M | 28.02 | 160.46 |
| WLP | 2.7B | 24.9M | 29.42 | 82.14 |

Table 1: Data statistics of three pre-training tasks. #query tokens represent average number of tokens per query, and #doc tokens represent average number of tokens per passage.

# 4 EXPERIMENTS

## 4.1 EXPERIMENTAL SETTING

**The two-tower retrieval model**   Each tower of the retrieval model follows the architecture and hyper-parameters of the 12 layers BERT-base model. For both towers, the final embedding is generated by applying a linear layer on the hidden state of the [CLS] token. The embedding dimension is 512. The sequence length for the query encoder and document encoder are set to be 64 and 288, respectively. We pre-train the model on 32 TPU v3 chips for 100K steps with an Adam optimizer and batch size of 8192. This process takes about 2.5 days. We use the Adam optimizer with an initial learning rate $1 \times 10^{-4}$ with the warm-up ratio 0.1, followed by a linear learning rate decay. For fine-tuning, the learning rate of Adam is set to $5 \times 10^{-5}$ with 2000 training steps and batch size 512.

**Pre-training tasks**   We compare the token-level pre-training task MLM with the three paragraph-level pre-training tasks, ICT, BFS and WLP. The data of ICT, BFS and WLP are generated from the Wikipedia corpus. The data statistics are reported in Table 1. Note that #tokens represents the number of sub-words tokenized by WordPiece (Wu et al., 2016). The pre-training tasks define the positive $(q, d)$ pair for learning the two-tower Transformer models. For ICT, the $d$ is a pair of article title and passage separated by [SEP] symbol as input to the doc-tower.

We propose to pre-train the two-tower Transformer models jointly with all three paragraph-level pre-training tasks, hence the name ICT+BFS+WLP. Here the model is pre-trained on one combined set of $(q, d)$ pairs, where each pair is uniformly sampled from the three pre-training tasks in Table 1. See Section 4.2 and 4.3 for its outstanding performance over other baselines.

**Downstream tasks**   We consider the Retrieval Question-Answering (ReQA) benchmark, proposed by Ahmad et al. (2019).[2] The two QA datasets we consider are SQuAD and Natural Questions. Note that each entry of QA datasets is a tuple $(q, a, p)$, where $q$ is the question, $a$ is the answer span, and $p$ is the evidence passage containing $a$. Following  Ahmad et al. (2019), we split a passage into sentences, $p = s_1 s_2 \ldots s_n$ and transform the original entry $(q, a, p)$ to a new tuple $(q, s_i, p)$ where $s_i$ is the sentence contains the answer span $a$.

The retrieval problem is that given a question $q$, retrieve the correct sentence and evidence passage pair $(s, p)$ from all candidates. For each passage $p$, we create a set of candidate pairs $(s_i, p)$ where $i = 1 \ldots n$, and the retrieval candidate set is built by combining such pairs for all passages. This problem is more challenging than retrieving the evidence passage only since the larger number of candidates to be retrieved. The data statistics of the downstream ReQA benchmark are shown in Table 2. Note that, similar to  Ahmad et al. (2019), the ReQA benchmark is not entirely open-domain QA retrieval as the candidates $(s, p)$ only cover the training set of QA dataset instead of entire Wikipedia articles. For the open-domain retrieval experiment, see details in Section 4.4.

**Evaluation**   For each dataset, we consider different training/test split of the data ($1\%/99\%$, $5\%/95\%$ and, $80\%/20\%$) in the fine-tuning stage and the 10% of training set is held out as the validation set for hyper-parameter tuning. The split is created assuming a cold-start retrieval scenario where the queries in the test (query, document) pairs are not seen in training.

---

[2]Different from (Ahmad et al., 2019), whose goal is to use other large-scale weakly-supervised query-answer pair datasets (e.g. reddit data) to improve the model, the goal of this paper is to study different unsupervised pre-training tasks not identical to the downstream task. Therefore our approaches are not directly comparable to the results presented in their paper.

| ReQA Dataset | #query | #candidate | #tuples | #query tokens | #doc tokens |
|---|---|---|---|---|---|
| SQuAD | 97,888 | 101,951 | 99,024 | 11.55 | 291.35 |
| Natural Questions | 74,097 | 239,008 | 74,097 | 9.29 | 352.67 |

Table 2: Data statistics of ReQA benchmark. candidate represents all (sentence, passage) pairs.

| train/test ratio | Encoder | Pre-training task | R@1 | R@5 | R@10 | R@50 | R@100 |
|---|---|---|---|---|---|---|---|
| 1%/99% | BM-25 | No Pretraining | **41.86** | 58.00 | 63.64 | 74.15 | 77.91 |
| | BoW-MLP | No Pretraining | 0.14 | 0.35 | 0.49 | 1.13 | 1.72 |
| | BoW-MLP | ICT+BFS+WLP | 22.55 | 41.03 | 49.93 | 69.70 | 77.01 |
| | Transformer | No Pretraining | 0.02 | 0.06 | 0.08 | 0.31 | 0.54 |
| | Transformer | MLM | 0.18 | 0.51 | 0.82 | 2.46 | 3.93 |
| | Transformer | ICT+BFS+WLP | 37.43 | **61.48** | **70.18** | **85.37** | **89.85** |
| 5%/95% | BM-25 | No Pretraining | 41.87 | 57.98 | 63.63 | 74.17 | 77.91 |
| | BoW-MLP | No Pretraining | 1.13 | 2.68 | 3.62 | 7.16 | 9.55 |
| | BoW-MLP | ICT+BFS+WLP | 26.23 | 46.49 | 55.68 | 75.28 | 81.89 |
| | Transformer | No Pretraining | 0.17 | 0.36 | 0.54 | 1.43 | 2.17 |
| | Transformer | MLM | 1.19 | 3.59 | 5.40 | 12.52 | 17.41 |
| | Transformer | ICT+BFS+WLP | **45.90** | **70.89** | **78.47** | **90.49** | **93.64** |
| 80%/20% | BM-25 | No Pretraining | 41.77 | 57.95 | 63.55 | 73.94 | 77.49 |
| | BoW-MLP | No Pretraining | 19.65 | 36.31 | 44.19 | 62.40 | 69.19 |
| | BoW-MLP | ICT+BFS+WLP | 32.24 | 55.26 | 65.49 | 83.37 | 88.50 |
| | Transformer | No Pretraining | 12.32 | 26.88 | 34.46 | 53.74 | 61.53 |
| | Transformer | MLM | 27.34 | 49.59 | 58.17 | 74.89 | 80.33 |
| | Transformer | ICT+BFS+WLP | **58.35** | **82.76** | **88.44** | **95.87** | **97.49** |

Table 3: Recall@k on SQuAD. Numbers are in percentage (%).

For the evaluation metric, we focus on recall@k[3] because the goal of the retrieval phase is to capture the positives in the top-k results. The retrieval performance can be understood independently of the scoring model used by measuring recall at different k. In fact, in the extreme cases when the scoring model is either oracle or random, the final precision metric is proportional to recall@k.

## 4.2 MAIN RESULTS

Table 3 and Table 4 compare the proposed combination of pre-training tasks, ICT+BFS+WLP, to various baselines on SQuAD and Natural Questions, respectively. In both benchmarks, ICT+BFS+WLP notably outperforms all other methods. This suggests that *one should use a two-tower Transformer model with properly designed pre-training tasks in the retrieval stage to replace the widely used* BM-25 *algorithm.* We present some of the detailed findings below.

**The BM-25 baseline** In retrieval, BM-25 is a simple but tough-to-beat unsupervised baseline using token-matching with TF-IDF weights as the scoring function. BM-25 performs especially well for the SQuAD benchmark, as the data collection process and human annotations of this dataset are biased towards question-answer pairs with overlapping tokens (Rajpurkar et al., 2016; Kwiatkowski et al., 2019). For instance, in the limited fine-tuning data scenario (e.g., 1% and 5%), BM-25 outperforms the two-tower transformer models with no pre-training (No Pretraining) or with less-effective pre-training tasks (MLM). This result verifies that BM-25 is a robust retrieval model and therefore widely used in recent works (Chen et al., 2017; Yang et al., 2017; Lee et al., 2019)[4].

---

[3]The correctness is based on when the system retrieves the gold sentence and evidence paragraph pair , not just any paragraph containing the answer text.

[4]Our BM-25 results are consistent with Ahmad et al. (2019). Their numbers are slightly higher because they consider passage-level retrieval, which has smaller candidate set compared to our sentence-level retrieval.

**Encoder architecture**   We justify the use of Transformer as encoders by comparing it with a shallow bag-of-word MLP model (BoW-MLP). Specifically, BoW-MLP looks up uni-grams from the embedding table[5], aggregates the embeddings with average pooling, and passes them through a shallow two-layer MLP network with tanh activation to generate the final 512-dimensional query/document embeddings. For fair comparison, the BoW-MLP encoder has a comparable model size to the Transformer encoder (i.e., 128M v.s. 110M parameters, slightly favorable to BoW-MLP encoder).

With a properly designed pre-training task (e.g., ICT+BFS+WLP), the Transformer encoder considerably outperforms its shallow counterpart (BoW-MLP), suggesting that the former benefits more from the unsupervised pre-training tasks. On the other hand, without any pre-training, the performance of the Transformer encoder is worse than BoW-MLP encoder, possibly because the former is over-fitting on the limited amount of labeled fine-tuning data.

**Pre-training tasks**   When pre-training the two-tower Transformer model, we compare the pre-training tasks to two baselines: No Pretraining and MLM. No Pretraining represents random initializing the model, and MLM is the token-level masked-LM task introduced in Section 3.

On both datasets, the token-level pre-training task MLM only marginally improves over the no-pretraining baseline (No Pretraining). In contrast, combining the paragraph-level pre-training tasks ICT+BFS+WLP provides a huge boost on the performance. This verifies our assumption that the design of task-related pre-training tasks is crucial. The performance of adding individual pre-training tasks is presented in the next section.

| train/test ratio | Encoder | Pre-training task | R@1 | R@5 | R@10 | R@50 | R@100 |
|---|---|---|---|---|---|---|---|
| 1%/99% | BM-25 | No Pretraining | 4.99 | 11.91 | 15.41 | 24.00 | 27.97 |
| | BoW-MLP | No Pretraining | 0.28 | 0.80 | 1.08 | 2.02 | 2.66 |
| | BoW-MLP | ICT+BFS+WLP | 9.22 | 24.98 | 33.36 | 53.67 | 61.30 |
| | Transformer | No Pretraining | 0.07 | 0.19 | 0.28 | 0.56 | 0.85 |
| | Transformer | MLM | 0.18 | 0.56 | 0.81 | 1.95 | 2.98 |
| | Transformer | ICT+BFS+WLP | **17.31** | **43.62** | **55.00** | **76.59** | **82.84** |
| 5%/95% | BM-25 | No Pretraining | 5.03 | 11.96 | 15.47 | 24.04 | 28.00 |
| | BoW-MLP | No Pretraining | 1.36 | 3.77 | 4.98 | 8.56 | 10.77 |
| | BoW-MLP | ICT+BFS+WLP | 11.40 | 30.64 | 40.63 | 62.95 | 70.85 |
| | Transformer | No Pretraining | 0.37 | 1.07 | 1.40 | 2.73 | 3.82 |
| | Transformer | MLM | 1.10 | 3.42 | 4.89 | 10.49 | 14.37 |
| | Transformer | ICT+BFS+WLP | **21.46** | **51.03** | **62.99** | **83.04** | **88.05** |
| 80%/20% | BM-25 | No Pretraining | 4.93 | 11.52 | 14.96 | 23.64 | 27.77 |
| | BoW-MLP | No Pretraining | 9.78 | 26.76 | 34.16 | 50.34 | 56.44 |
| | BoW-MLP | ICT+BFS+WLP | 13.58 | 37.78 | 50.40 | 76.11 | 82.98 |
| | Transformer | No Pretraining | 7.49 | 20.11 | 25.40 | 38.26 | 43.75 |
| | Transformer | MLM | 16.74 | 40.48 | 49.53 | 67.91 | 73.91 |
| | Transformer | ICT+BFS+WLP | **30.27** | **63.97** | **75.85** | **91.84** | **94.60** |

Table 4: Recall@k on Natural Questions. Numbers are in percentage (%).

## 4.3   ABLATION STUDY

We conduct a more thorough ablation study on Natural Questions involving (1) the number of layers in Transformer; (2) different pre-training tasks; and (3) dimension of the embedding space. The result is presented in Table 5.

Index 1, 2, and 3 show the individual performance of three pre-training tasks. All of these tasks are much more effective than MLM. Among them, ICT has the best performance, followed by BFS, and then WLP. This suggests that the (query, document) pairs defined by local context within passage are suitable for the ReQA task.

[5]We empirically found that adding bi-grams does not further improve the performance on these tasks possibly due to over-fitting.

| Index | Ablation Configuration | | | R@100 on different train/test ratio | | | |
|:---:|:---:|:---:|:---:|:---:|:---:|:---:|:---:|
| | #layer | Pre-training task | emb-dim | 1% | 5% | 10% | 80% |
| 1 | 4 | ICT | 128 | 77.13 | 82.03 | 84.22 | 91.88 |
| 2 | 4 | BFS | 128 | 72.99 | 78.34 | 80.47 | 89.82 |
| 3 | 4 | WLP | 128 | 56.94 | 68.08 | 72.51 | 86.15 |
| 4 | 12 | No Pretraining | 128 | 0.72 | 3.88 | 6.94 | 38.94 |
| 5 | 12 | MLM | 128 | 2.99 | 12.21 | 22.97 | 71.12 |
| 6 | 12 | ICT | 128 | 79.80 | 85.97 | 88.13 | 93.91 |
| 7 | 12 | ICT+BFS+WLP | 128 | 81.31 | 87.08 | 89.06 | 94.37 |
| 8 | 12 | ICT+BFS+WLP | 256 | 81.48 | 87.74 | 89.54 | 94.73 |
| 9 | 12 | ICT+BFS+WLP | 512 | 82.84 | 88.05 | 90.03 | 94.60 |

Table 5: Ablation study on Natural Questions based on Recall@100. Index 9 represents the proposed method in Table 4.

Also note from Index 6 and 7, ICT+BFS+WLP pre-training is better than ICT with 1.5% absolute improvement over ICT in the low-data regime. This reflects that, when there's no sufficient downstream training data, more globally pre-training tasks is beneficial as it encodes multi-hop reasoning priors such as different passages within the same article (BFS) or even going beyond to different articles linked by the same entities (WLP).

Finally, The advantage of increasing number of layers is manifest by comparing Index 1 and Index 6, while Index 7, 8 and 9 show the benefit of increasing the dimension of the embedding space.

## 4.4 EVALUATION OF OPEN-DOMAIN RETRIEVAL

We consider the open-domain retrieval setting by augmenting the candidate set of the ReQA benchmark with large-scale (sentence, evidence passage) pairs extracted from general Wikipedia articles. In particular, we preprocess/sub-sample the open-domain Wikipedia retrieval set of the DrQA paper (Chen et al., 2017) into one million (sentence, evidence passage) pairs, and add this external 1M candidate pairs into the existing retrieval candidate set of the ReQA benchmark.

| train/test ratio | Pre-training task | R@1 | R@5 | R@10 | R@50 | R@100 |
|:---:|:---:|:---:|:---:|:---:|:---:|:---:|
| | BM-25 | 3.70 | 9.58 | 12.69 | 20.27 | 23.83 |
| 1%/99% | ICT | **14.18** | 37.36 | 48.08 | 69.23 | 76.01 |
| | ICT+BFS+WLP | 13.19 | **37.61** | **48.77** | **70.43** | **77.20** |
| | BM-25 | 3.21 | 8.62 | 11.50 | 18.59 | 21.78 |
| 5%/95% | ICT | **17.94** | 45.65 | 57.11 | 76.87 | 82.60 |
| | ICT+BFS+WLP | 17.62 | **45.92** | **57.75** | **78.14** | **83.78** |
| | BM-25 | 3.12 | 8.45 | 11.18 | 18.05 | 21.30 |
| 80%/20% | ICT | 24.89 | 57.89 | 69.86 | 87.67 | 91.29 |
| | ICT+BFS+WLP | **25.41** | **59.36** | **71.12** | **88.25** | **91.71** |

Table 6: Open-domain retrieval results of Natural Questions dataset, where existing candidates are augmented with additional 1M retrieval candidates (i.e., 1M of $(s, p)$ candidate pairs) extracted from open-domain Wikipedia articles.

The results of open-domain retrieval on Natural Questions are presented in Table 6. Firstly, we see that the two-tower Transformer models pretrained with ICT+BFS+WLP and ICT substantially outperform the BM-25 baseline. Secondly, ICT+BFS+WLP pre-training method consistently improves the ICT pre-training method in most cases. Interestingly, the improvements are more noticeable at R@50 and R@100, possibly due to that the distant multi-hop per-training supervision induces better retrieval quality at the latter part of the rank list. Finally, we conclude that the evaluation results of the 1M open-domain retrieval are consistent with our previous empirical evaluation on the ReQA benchmark with smaller retrieval candidate sets (Section 4.2).

## 5 CONCLUSION

We conducted a comprehensive study on how various pre-training tasks help in the large-scale retrieval problem such as evidence retrieval for question-answering. We showed that the two-tower Transformer models with random initialization (No Pretraining) or the unsuitable token-level pre-training task (MLM) are no better than the robust IR baseline BM-25 in most cases. With properly designed paragraph-level pre-training tasks inlcuding ICT, BFS and WLP, the two-tower Transformer models can considerably improve over the widely used BM-25 algorithm.

For future works, we plan to study how the pre-training tasks apply to other types of encoders architectures, generating the pre-training data from corpora other than Wikipedia, and how pre-training compares with different types of regularizations.

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
