# OpenReview forum: "Pre-training Tasks for Embedding-based Large-scale Retrieval"
_ICLR.cc/2020/Conference — Accept (Poster)_

### Official Review · AnonReviewer2 · 2019-10-20
**Official Blind Review #2**

**Rating:** 6

**Review:**

This paper studies a query-related document retrieval problem using a framework which they call “two-tower retrieval method”. The task is to learn query representation and document representation in order to retrieve query-related documents by the maximum inner product. This is a realistic setting for large-scale retrieval problem since it enables document representations to be computed once regardless of the question, and obtaining query-sensitive document representations is very expensive.
Then, the paper studies three different pretraining methods for this task, ICT (previously proposed by Lee et al 2019), and BFS & WLP (proposed by this paper).
For evaluation, the paper considers the retrieval task of question answering, based on SQuAD and Natural Questions. The combination of ICT, BFS and WLP achieves remarkable improvement over the number of baselines including BM25 and other neural-based models.

The strength of this paper is that it includes comprehensive studies on the two-tower retrieval problem. In particular, they have conducted extensive ablation studies with different train/test ratios.

However, there are some notable weaknesses of this paper as follows.

First, the benchmark relies on the recall rate instead of the end task (open-domain QA). Recall rate is not a good way to evaluate the retrieval result since a system may retrieve text which contains the answer text but is not semantically related to the question. (I understand that the paper follows Admad et al (2019), but I believe this is not a published paper.) In addition, this paper did not empirically demonstrate the relatedness between the recall rate and the end performance. This makes it very hard to compare with other papers in open-domain QA, which has been extensively studied for a few recent years.

Second, despite comprehensive studies, the fact that ICT+BFS+WLP is almost the same as ICT (93.91 vs. 94.37) means that the method does not give improvement over ICT which was already proposed in the previous study.

Third, the gap between BM25 and ICT+BFS+WLP in Table 3 and 4 are very significant (e.g. 27 vs 94 on Natural Questions), but this doesn't seem to be consistent to Lee et al (2019). (There are some differences: (1) Lee et al (2019) compares BM25 vs. ICT, but according to this paper, ICT and ICT+BFS+WLP are similar. (2) Lee et al (2019) reports the end QA performance while this paper reports the recall rate, but one of the assumptions in this paper is that recall rate and the end performance is related.) What is the explanation for this discrepancy?

(I am happy to increase the rating if my concerns are resolved during rebuttals and/or the paper includes performance on the end QA performance.)

Some questions:
1) Section 4.1 says ICT is sentence-level, BFS is paragraph-level and WLP is document-level. What does it mean? I thought, according to Section 3, all methods are paragraph-level.
2) Section 4.1: it looks like Ahmad et al (2019)’s setting is actually not entirely open-domain. Their candidate sentences/paragraphs are much less than the entire Wikipedia. Did this paper also use the same set of the candidate? In that case, it should be clearly mentioned in the paper. In addition, the data statistics are different across two papers. Did Ahmad et al (2019) include only train set whereas this submission reports train+test? In case there is an official split of train/test, why were different splits used for evaluation?
3) Also regarding the split: for each split, how much was used for development? I believe data used for the development and test should be different. In fact, rather than experimenting on different ratios of train/test, is it possible to report on official test set, while splitting the train set into 90/10 for training and development? Or, split the entire data to 90/5/5 for training/development/test?



Update on Nov 15: The revised paper resolves most of my concerns, so I am updating the score from 3 to 6.

**Experience Assessment:**

I have published one or two papers in this area.

**Review Assessment: Checking Correctness Of Derivations And Theory:**

I assessed the sensibility of the derivations and theory.

**Review Assessment: Checking Correctness Of Experiments:**

I carefully checked the experiments.

**Review Assessment: Thoroughness In Paper Reading:**

I read the paper at least twice and used my best judgement in assessing the paper.

---

> ### Author Response · Authors · 2019-11-07
> **Response to R2**
>
> We thank the reviewer for the detailed comments.
>
>
> Comment1:“First, the benchmark relies on the recall rate instead of the end task (open-domain QA),...”
>
> We focus on Recall@k in this paper because the goal of the retrieval phase is to capture the positives in the top-k results. The retrieval performance can be understood independently of the scoring model used by measuring recall at different k. In fact, in extreme cases when the scoring model is oracle or random, the final precision metric is proportional to recall@k.
>
> We believe recall@k is sufficient in this setup but we are also working on a new end-to-end trained retrieval+scoring setup as a followup work.
>
> When computing recall@k, we consider it as correct only when the system retrieves the gold evidence paragraph and sentence, not just any paragraph containing the answer text.
>
>
>
> Comment2: “ICT+BFS+WLP is almost the same as ICT (93.91 vs. 94.37),...”
>
> Indeed ICT is a very effective pre-training task as demonstrated in our experiments. But also note that, in the low-data regime of Table 5, we see a 1.5% absolute improvement of R@100 when comparing ICT+BFS+WLP to ICT. This reflects that, when there’s no sufficient downstream training data, more globally pre-training tasks is beneficial as it encodes multi-hop reasoning priors such as different passages within the same article (BFS) or even going beyond different articles linked by the same entities (WLP).
>
> The major contribution of this work is to provide a comprehensive study of different sentence-level pre-training tasks for the two-tower transformer-based retrieval models. We believe this paper provides a set of standard baselines that can be helpful for further explorations in the community.
>
>
>
> Comment3: “the gap between BM25 and ICT+BFS+WLP in Table 3 and 4 are very significant (e.g. 27 vs 94 on Natural Questions), but this doesn't seem to be consistent to Lee et al (2019).”
>
> (1) Lee et al (2019) evaluate on “exact match” metric of Open-Domain QA problems, and they did not finetune the doc-tower encoder on the QA problem. This indeed makes the comparison of their Table 5 difficult to compare with our results.
> (2) For both SQuad and NQ datasets, our BM-25 results are consistent with Ahmad et. al. 2019. See Table 6 of their paper for more details. Note that their numbers are slightly higher than ours because they evaluate passage-level retrieval, which has a smaller candidate set compared to our sentence-level retrieval (i.e. candidate c=(sent, passage)).
>
>
>
> Q1: “Section 4.1 says ICT is sentence-level, BFS is paragraph-level and WLP is document-level. What does it mean?”
>
> Thanks for pointing this out. We agree notating ICT, BFS, WLP this way is confusing. We will change the description in the next version.
>
>
> Q2: “it looks like Ahmad et al (2019)’s setting is actually not entirely open-domain… Did this paper also use the same set of the candidate? ...”
>
> Similar to Ahmad et. al. (2019), this paper is not entirely open-domain. The candidate sets are converted from the question/answer/evidence tuple in SQuad and Natural Questions datasets. We will make this clear in the updated version of the paper.
>
> Our paper uses a different split from Ahmad et. al. (2019) since their benchmark does not have a train/test split. They use the benchmark to test zero-shot performance while we are interested in studying pre-training under the pre-training/fine-tuning framework. We report results on various train/test split so that we can understand the effect of pre-training when the different amounts of fine-tuning data are available.
>
>
> Q3: “Also regarding the split: for each split, how much was used for development?”
>
> The development set is created by holding out 10% of the training set for hyper-parameter tuning. We will clarify this in the final version.
>
> We can add the suggested setting in the final version.

---

> > ### Comment · AnonReviewer2 · 2019-11-09
> > **Thanks for clarification**
> >
> > Thanks for your response.
> >
> > Regarding Comment1:“First, the benchmark relies on the recall rate instead of the end task (open-domain QA),...”
> > > I did not realize that the recall was based on the groundtruth paragraph, not just any paragraph containing the answer text. Sorry for missing this detail, and it resolves my concern.
> >
> > Regarding Comment2: “ICT+BFS+WLP is almost the same as ICT (93.91 vs. 94.37),...”
> > > I agree that there is a performance improvement when the train data is insufficient, but splitting train/test into 1%/99% or 5%/95% seems like a very unrealistic setting. Instead of that, how about evaluating on the dataset that has a small size, such as TREC or WebQuestions?
> >
> > Regarding Q2:
> > I think the fact that the setting is not totally open-domain but calling it open-domain can be very misleading. In particular, SQuAD only contains 500 distinct articles. For me, dealing with 500 articles and dealing with a web-scale corpus are totally different problems.
> > I wonder if authors can experiment on the real open-domain setting, as it doesn't seem to be much harder to experiment on but is much more realistic, and is easier to compare with previous studies.
> >
> > For other comments/questions: Responses from authors resolve my concerns, and I look forward to the updated version of the paper.

---

> > > ### Author Response · Authors · 2019-11-14
> > > **revision uploaded**
> > >
> > > Thanks for your reply. We have incorporated your suggestions in the uploaded version. In specific, we have conducted an open-domain retrieval experiment with 1M additional retrieval candidates in Section 4.4.

---

> > > > ### Comment · AnonReviewer2 · 2019-11-15
> > > > **Revised paper substantially resolves my concerns.**
> > > >
> > > > Hi, thanks for updating the paper. I think reviewed paper substantially resolves my concerns, especially with some clarification about the data & settings, and the new results in an open-domain setting. I am still curious why BM25 performance is very bad, which is quite opposite to some previous studies, but I guess possibly BM25 is good for paragraph-level retrieval but not for sentence-level retrieval.
> > > >
> > > > I increase the score from 3 to 6.

---

### Official Review · AnonReviewer1 · 2019-10-22
**Official Blind Review #1**

**Rating:** 6

**Review:**

The paper provides a comprehensive study on the two-tower Transformer models in terms of the impact of its pre-training tasks on large-scale retrieval applications. The studies here show that, pre-training with Inverse Cloze Task (ICT) the two-tower Transformer models significantly outperform the widely used BM-25 algorithm for large-scale information retrieval. The authors also propose two novel pre-training settings which also show improvement over the baseline BM-25. In addition, the authors empirically demonstrate that the token-level masked-LM model used by BERT is not a good choice as pre-training task for the two-tower Transformer when deployed for large-scale information retrieval applications.

The paper is well written and easy to follow. The Ablation Study of the paper also provides useful insights about the impact of different pre-training schemas on large-scale information retrieval tasks. I think the studies here will benefit the communities where large-scale information retrieval is required such as open-domain question answering.

The main limitation to me is that, the two novel pre-training tasks proposed in this paper are specific for Wikipedia and they are less effective than the ICT strategy (as shown in Table 5).

I hope the authors will release the source codes to the community.


**Experience Assessment:**

I have read many papers in this area.

**Review Assessment: Checking Correctness Of Derivations And Theory:**

I carefully checked the derivations and theory.

**Review Assessment: Checking Correctness Of Experiments:**

I carefully checked the experiments.

**Review Assessment: Thoroughness In Paper Reading:**

I read the paper thoroughly.

---

> ### Author Response · Authors · 2019-11-07
> **Response to R1**
>
> We thank the reviewer for the comments.
>
> The major contribution of this work is to provide a first comprehensive study of different sentence-level pre-training tasks for the two-tower transformer-based retrieval models. We believe this paper provides a set of standard baselines that can be helpful for further explorations in the community. Indeed one of our major findings is that ICT is a very effective sentence-level pre-training task, whereas the masked-LM only provides marginal improvements.
>
> In addition, for the low-data regime of Table 5, we see a 1.5% absolute improvement of R@100 when comparing ICT+BFS+WLP to ICT. This reflects that, when there’s no sufficient downstream training data,  more globally pre-training tasks is beneficial as it encodes multi-hop reasoning priors such as different passages within the same article (BFS) or even going beyond different articles linked by the same entities (WLP). We will add a discussion in the final version.
>
> We are working to release our experiment code, pre-trained two-tower transformer models, and downstream evaluation data benchmark.

---

> > ### Comment · AnonReviewer1 · 2019-11-15
> > **Thank you for your feedback**
> >
> > Thank you for your clarification; really appreciate that.
> >
> > I like your aim of providing a comprehensive study here and the take-home message seems clear to me. On the other hand, I do have concern on the limited novelty as mentioned above.

---

### Official Review · AnonReviewer3 · 2019-10-28
**Official Blind Review #3**

**Rating:** 1

**Review:**

This paper proposes a solution to the large scale query-document retrieval problem. The proposed method was shown to be a better alternative to the classic information retrieval approach such as BM-25 (token marching + TF-IDF weights). The proposed method is based on two separate transformer models which has computational benefit over one cross-attention model. For fast training, they have also used the sampled softmax. For pre-training tasks, Inverse Cloze Task (ICT), Body First Selection (BFS), Wiki LinkPrediction (WLP) were studied.


The paper is written well, easy to follow and well-motivated. However, there is a major technical problem in the proposed method. In the proposed approach, the query embedding (q_emb) and the document embedding (d_emb) train separately by two transformer models (two towers --- Query-tower and Doc-tower). After that, the similarity was measured through a dot product. Two embedding models are, therefore, represented by separate vector space representation. Applying dot product to find the similarity does not make much sense to me, as the embedding is not comparable in two different vector representations.


**Experience Assessment:**

I have published one or two papers in this area.

**Review Assessment: Checking Correctness Of Derivations And Theory:**

I assessed the sensibility of the derivations and theory.

**Review Assessment: Checking Correctness Of Experiments:**

I assessed the sensibility of the experiments.

**Review Assessment: Thoroughness In Paper Reading:**

I read the paper thoroughly.

---

> ### Author Response · Authors · 2019-11-07
> **Response to R3**
>
> We thank the reviewer for the comments.
>
> The reviewer misunderstood the paper. Both the query tower and doc tower are trained jointly end-to-end -- they are embedding the queries and docs in the same embedding space, where the dot product is used to measure similarity.
>
> In fact two-tower models like this are widely used in question answering [1,2], information retrieval [3,4], dialogue [5,6] etc.
>
> [1] Cer et al. Universal Sentence Encoder. ACL 2018
> [2] Das et al. Together we stand: Siamese networks for similar question retrieval. ACL 2016
> [3] Krichene et al. Efficient Training on Very Large Corpora via Gramian Estimation. ICLR 2019
> [4] Wu et al. StarSpace: Embed All The Things! AAAI 2018
> [5] Zhang et al.  Personalizing dialogue agents: I have a dog, do you have pets too. ACL 2018
> [6] Mazaré et al. Training millions of personalized dialogue agents. EMNLP 2018

---

### Author Response · Authors · 2019-11-14
**Revision Uploaded**

To all reviewers,

Thanks again for all your constructive reviews. We have uploaded a revised version of our manuscript, with the differences highlighted as follows.

(1) Rephrased the ICT, BFS, and WLP as paragraph-level pre-training tasks for consistency.

(2) On page 6, the Downstream paragraph, clarified that our ReQA benchmark is not entirely open-domain retrieval (candidates are not constructed from entirely Wikipedia articles).

(3) On page 6, the Evaluation paragraph, clarified why we use evaluation metric R@k and how we compute it. We also explained the train/test data split, where the validation is an additional split from the training set.

(4) On page 7, the BM25 paragraph, explained that the BM25 baseline results are consistent with the ReQA paper (Ahmad et al. 2019).

(5) On page 8, the third paragraph, clarified the proposed ICT+BFS+WLP method has 1.5 absolute improvement over ICT in the low-data regime and provided further explanations.

(6) On page 9, Section 4.4, conducted the open-domain retrieval evaluation, which has 1M additional retrieval candidates. See more details in Table 6 and Section 4.4.

---

### Decision · Program_Chairs · 2019-12-19

**Decision:**

Accept (Poster)

**Comment:**

This paper conducts a comprehensive study on different retrieval algorithms and show that the two-tower Transformer models with properly designed pre-training tasks can largely improve over the widely used BM-25 algorithm. In fact, the deep learning based two tower retrieval model is already used in the IR field. The main contribution lies in the comprehensive experimental evaluation.

Blind Review #3 has a major misunderstanding of the paper; hence his review will be excluded. The other two reviewers tend to accept the paper with several minor comments.

As the authors promise to release the code as a baseline for further works, I agree to accept the paper.